# Cascadia low frequency earthquakes at the base of an overpressured subduction shear zone

Andrew J. Calvert[1✉], Michael G. Bostock[2], Geneviève Savard[3] & Martyn J. Unsworth[4]

In subduction zones, landward dipping regions of low shear wave velocity and elevated Poisson's ratio, which can extend to at least 120 km depth, are interpreted to be all or part of the subducting igneous oceanic crust. This crust is considered to be overpressured, because fluids within it are trapped beneath an impermeable seal along the overlying inter-plate boundary. Here we show that during slow slip on the plate boundary beneath southern Vancouver Island, low frequency earthquakes occur immediately below both the landward dipping region of high Poisson's ratio and a 6–10 km thick shear zone revealed by seismic reflections. The plate boundary here either corresponds to the low frequency earthquakes or to the anomalous elastic properties in the lower 3–5 km of the shear zone immediately above them. This zone of high Poisson's ratio, which approximately coincides with an electrically conductive layer, can be explained by slab-derived fluids trapped at near-lithostatic pore pressures.

[1] Department of Earth Sciences, Simon Fraser University, 8888 University Drive, Burnaby, BC V5A 1S6, Canada. [2] Department of Earth, Ocean and Atmospheric Sciences, 2207 Main Mall, University of British Columbia, Vancouver, BC V6T 1Z4, Canada. [3] Department of Geosciences, University of Calgary, 2500 University Drive NW, Calgary, AB T2N 1N4, Canada. [4] Department of Physics, University of Alberta, Edmonton, AB T6G 2E9, Canada. ✉email: acalvert@sfu.ca

To understand the tectonic processes and great seismic hazard of subduction zones, it is necessary to accurately locate the inter-plate boundary, below which an oceanic plate is thrust down into the mantle. The inter-plate boundary is defined by the occurrence of slip and includes a locked seismogenic zone that can extend close to the offshore trench or deformation front, a deeper section where episodic slow slip may occur, and the deepest section where there is stable sliding[1,2]. Slip on the plate boundary is revealed by large magnitude earthquakes on a low-angle thrust fault and/or by deformation that can be measured at Earth's surface. Megathrust earthquakes occur in the shallower locked zone, which is typically above 30 km depth[3], but in the northern Cascadia subduction zone where the mean recurrence interval is ~500–530 years[4], there is very little instrumentally recorded seismicity on the plate boundary[5]. At greater depths, low-frequency earthquakes[6] (LFEs), which are characterized on average by low-angle thrusting[7,8], and non-volcanic tremor occur during episodes of slow slip, and have the potential to define the location of the inter-plate boundary, but determining the hypocentres of such events is difficult due to their low magnitudes. Inter-seismic deformation at the surface produced during slow slip can be measured, for example using GPS (Global Positioning System) monitoring, but is relatively insensitive to the depth of the slip, and can only constrain the approximate geometry of the plate boundary. Thus it is often challenging to determine the position of the megathrust, and to locate relative to this boundary structures and processes inferred from geophysical observations. In the Cascadia subduction zone, for example, there are discrepancies up to 10 km in the interpreted depth of the inter-plate boundary in the region of slow slip[9,10], raising uncertainty about whether some features lie in the subducting or overriding plate.

Close to the deformation front, where its position can be verified by drilling, the inter-plate boundary can sometimes be identified in seismic reflection images[11] or, more commonly, inferred indirectly from underthrust sedimentary strata, but interpretation of such images becomes increasingly uncertain below the variable folds and faulting of the overriding plate and as plate-boundary-related reflectors develop into more complex structures with increasing depth. In Cascadia, the megathrust fault appears to be a relatively thin, <2 km, reflector in the seismogenic zone that thickens landward into a package of reflectors up to 10 km thick[12] in the region of slow slip[13]; these observations have led to the proposal that this transition indicates a landward change to ductile deformation that is distributed vertically over the thick reflective zone[13]. A similar thickening of reflectors related to the plate boundary has also been found in the Alaska subduction zone[14]. In both these subduction zones, however, the lack of nearby, well-located thrust earthquakes means that the association of these reflections with the inter-plate boundary is an unproven, though likely reasonable, assumption. In Alaska when the few, poorly located thrust earthquakes within 80 km are projected onto the seismic reflection profile they lie 2–10 km below the inferred megathrust reflector[14]. In contrast, in the Sumatra subduction zone where offset landward dipping reflectors correspond to the top of the faulted igneous oceanic crust, a deeper non-reflective inter-plate boundary is interpreted within the uppermost mantle[15]. Though seismic reflection data provide relatively high spatial resolution, identification of the plate boundary, particularly at the depths where slow slip occurs, is uncertain in the absence of well-located evidence of low-angle thrusting.

Locations of the inter-plate boundary at depths >20 km have also been proposed based on the migration of teleseismic receiver functions[16], and in several subduction zones the underlying low-velocity zones (LVZs) have been interpreted as the igneous oceanic crust[17–20], including internal stratification[21], with the downdip extent of the LVZ upper boundary consistent with thermal-petrological modelling of the basalt-eclogite transition[17]. In Cascadia, detection of an approximately coincident 3–5 km thick, landward dipping zone with anomalously high Poisson's ratio of 0.3–0.4 has led to the proposal that the upper oceanic crust is maintained at near-lithostatic pore pressure by a low-permeability inter-plate boundary immediately above the LVZ[21,22], and a similar interpretation has been made in Japan[23,24]. In Cascadia, however, interpretations of the inter-plate boundary using receiver functions derived from teleseismic data are systematically shallower than interpretations based on active source, e.g., normal-incidence reflection and wide-angle, seismic surveys[9,10].

To reconcile the results from different seismic methods, it is necessary that they are presented in a common reference frame, i.e., depth. Conversion to depth of seismic recordings that are made in time requires the use of consistent P wave and S wave velocity models, but different seismic methods typically employ different models, contributing to discrepancies between the various studies. New 3D P and S wave velocity models have recently been developed for southern Vancouver Island in the northern Cascadia subduction zone by double-difference tomography, in which the local seismicity and LFEs are also relocated[25]. Using these models, we compare different results from active and passive seismic surveys previously acquired in the area. We show that relocated LFEs, which directly indicate part of the inter-plate boundary, lie immediately below a regionally extensive shear zone that includes the landward dipping zone of elevated Poisson's ratio previously interpreted to be metamorphosed subducting sediments[26] or overpressured upper oceanic crust[22] of the subducting Juan de Fuca plate. Consequently, the region of elevated Poisson's ratio previously associated with the subducting oceanic plate either lies within the overriding plate or forms part of a plate boundary zone a few km thick immediately above, and including, the LFEs. We also present new inversions of magnetotelluric (MT) data, showing that a conductivity anomaly consistent with increased fluid-filled porosity exists close to the plate boundary in this area. The downdip limit of this fluid saturated zone, which also rises into the North American plate, is marked by the landward termination of both the conductor and the zone of high Poisson's ratio. We suggest that the available geophysical data are consistent with both a thin inter-plate boundary and a vertically distributed inter-plate boundary in the zone of slow slip. We favour the model with a vertically distributed boundary, because active slip is likely able to generate and maintain the 3-5 km thick zone of anomalous elastic properties that are observed.

## Results

**Geophysical surveys of southern Vancouver Island.** To compare estimates of the inter-plate boundary from different seismic methods, we focus on southern Vancouver Island where teleseismic earthquakes have been recorded by seismograph stations distributed along the POLARIS (Portable Observatories for Lithospheric Analysis and Research Investigating Seismicity) profile[27] (Fig. 1); we use here a profile through the seismograph stations along an azimuth of 80°, rather than a profile perpendicular to the strike of the margin. Southern Vancouver Island also lies in the areas of investigation of the LITHOPROBE project[28] and the 1998 SHIPS (Seismic Hazards Investigation in Puget Sound) survey, which acquired marine seismic reflection data in the Strait of Juan de Fuca, Strait of Georgia and Puget Sound that were recorded at wide-angle by onshore stations and ocean-bottom seismometers[29–31]. Travel times from local earthquakes and LFEs recorded by all post-2002 seismograph stations, together with first arrivals from the SHIPS survey have been used in a double-difference tomographic inversion (TomoDD) to

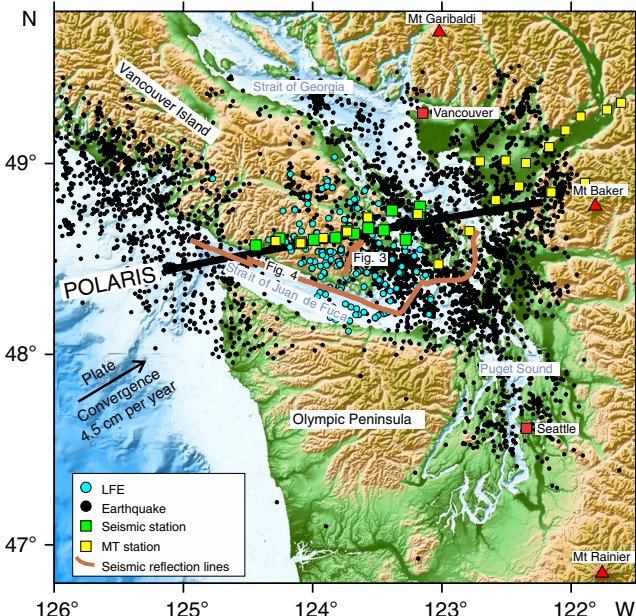

**Fig. 1 Location of geophysical studies in northern Cascadia subduction zone.** Solid black line shows the POLARIS (Portable Observatories for Lithospheric Analysis and Research Investigating Seismicity) profile along which the different sections in Fig. 2 were obtained from the tomographic velocity models. Solid brown lines are the short onshore and longer offshore seismic reflection sections shown in Figs. 3 and 4 respectively. The earthquakes and low-frequency earthquakes (LFE) used in the tomography study are shown by filled black and filled blue circles respectively. Green squares—seismic stations used to infer landward dipping zone of very high Poisson's ratio and ultra-low S wave velocity (ULVZ), yellow squares— magnetotelluric (MT) stations used to create resistivity section in Fig. 4.

develop consistent 3D P wave and S wave velocity models for the area[25] (see "Methods" section); relocated hypocentres were obtained for both the local earthquakes and LFEs. Long-period MT recordings have also been made close to the POLARIS profile as part of a larger study of the southern Canadian Cordillera[32,33], and a subset of these data have been reanalysed to provide complementary constraints on the variation of resistivity in the subsurface around southern Vancouver Island (see "Methods" section).

**Seismic images.** The seismic image along the POLARIS profile constructed by migration of teleseismic phases shows a prominent landward dipping S wave LVZ[27] (outlined by a dashed grey line in Fig. 2d), generally similar to those found in several other sub-duction zones[17–20]. A landward dipping region of very high Poisson's ratio, interpreted to represent an ultra-low S wave velo-city zone (ULVZ, between solid dark grey lines in Fig. 2d) [21,26], has also been identified here using the travel time differences of tele-seismic Ps and Pps phases[22], and mostly corresponds to the upper part of this LVZ. The landward dipping LVZ and ULVZ are superimposed on P wave and S wave velocity sections extracted from the 3-D velocity model along the POLARIS profile (Supplementary Fig. 1). To view spatial variation in the tomographic model more clearly, we have also superimposed these features on sections displaying functions of the Lamé elastic moduli, $\lambda$ and $\mu$:

$$\frac{\lambda}{\rho} = V_p^2 - 2V_s^2 \qquad (1)$$

$$\frac{\mu}{\rho} = V_s^2 \qquad (2)$$

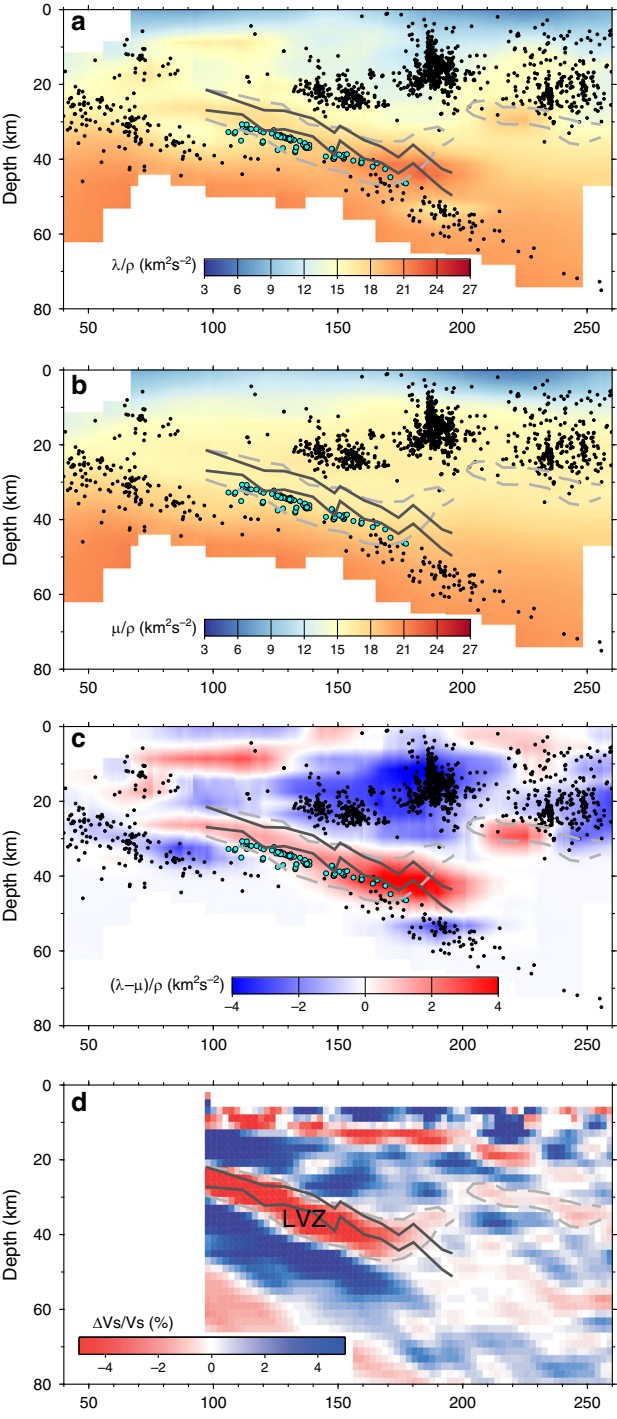

where $\rho$ is the mass density, $V_p$ is the P wave velocity, and $V_s$ is the S wave velocity (Fig. 2a, b). The LVZ, and a secondary lower amplitude feature in Fig. 2d, both outlined by dashed grey lines, correspond approximately to locally elevated values of $\lambda/\rho$ (and $V_p$), but no obvious features in $\mu/\rho$ (or $V_s$). Subtraction of $\mu/\rho$ from $\lambda/\rho$ reveals a landward dipping zone of anomalously high values, >2 km² s⁻² (Fig. 2c), which is where Poisson's ratio is elevated to 0.26–0.28 (Supplementary Fig. 1c). The strong correlation between the LVZ/ULVZ and the positive $(\lambda-\mu)/\rho$ and Poisson's ratio anomalies derived from the tomographic velocity models suggests that the tomographic and teleseismic methods are imaging the same subsurface structure, but in the tomographic inversion this feature exhibits a downward increase in P wave velocity rather than

**Fig. 2 Seismic sections along the POLARIS (Portable Observatories for Lithospheric Analysis and Research Investigating Seismicity) profile. a** Lamé modulus $\lambda/\rho$, where $\rho$ is density, showing locally elevated values at 42 km depth near the tip of the mantle wedge. **b** Lamé modulus $\mu/\rho$, which is equivalent to the square of the S wave velocity, showing a relatively smooth variation along the profile. **c** Difference $(\lambda-\mu)/\rho$, which exhibits a similar variation to Poisson's ratio (Supplementary Fig. 1c), **d** Migration of teleseismic receiver functions projected onto profile shown in Fig. 1. Landward dipping regions of high Poisson's ratio inferred from the relative travel times of teleseismic phases (solid dark grey lines) and negative S velocity perturbation inferred from teleseismic migration (grey dashed line) correspond to elevated values of $(\lambda-\mu)/\rho$ and Poisson's ratio in the tomographic velocity model. LVZ—low S wave velocity zone, LFEs—filled blue circles, earthquakes—filled black circles. Distance scale as in ref. [25]. Vertical exaggeration is 1.5.

the decrease in S wave velocity required by the teleseismic data. An S wave LVZ exists in the tomographic velocity model, but it is lower in magnitude, and dips more shallowly than in the teleseismic image, extending into the lower continental crust at 30–35 km depth, resulting in an increase in the tomographic S wave velocity through the lower part of the LVZ (Supplementary Fig. 1d).

The relocated local seismicity and LFEs within 20 km have been projected along strike onto the POLARIS profile using a local strike azimuth of 320°. Allowing for a typical ±2 km depth uncertainty[8,25], the LFEs (filled blue circles in Fig. 2) appear to be distributed close to a landward dipping surface and away from regular seismicity[25] (filled black circles in Fig. 2). Since the LFEs arise from low-angle thrusting to the northeast, based on double-couple moment tensor inversion of LFE templates[8], some, though perhaps a small proportion[34], of the slip on the inter-plate boundary must be accommodated across a thin zone coinciding with the LFEs.

Projection of the relocated LFEs within 20 km onto a short LITHOPROBE seismic reflection line adjacent to the POLARIS profile (Fig. 1) shows that the LFEs occur 0–3 km below the base of a ~6 km thick band of seismic reflectivity known as the E reflections (Fig. 3)[35]. Given the uncertainty in the hypocentre locations and the projection along strike some LFEs could lie within the deepest E reflectors. Projection of LFEs onto a composite seismic reflection section constructed from the SHIPS reflection lines around the southern end of Vancouver Island (Fig. 1) shows that here the LFEs also occur immediately below a 6–10 km thick band of seismic reflectivity (Fig. 4). These E reflections, which are notably aseismic, dip landward and extend to at least 50 km depth, which is well below the ~35 km Moho of the overriding plate[36]. Since overlying structures such as the terrane bounding Leech River Fault flatten into or are truncated by the top of the E reflectors, they have been previously interpreted as a shear zone[13,37], but whether individual reflectors arise from imbrication of different lithologies[38], mylonite zones, perhaps associated with the presence of fluid-filled porosity[13,37], or another mechanism is uncertain. At 30–35 km depth, P wave velocities in the E reflectors increase northward from ~6.6 km s$^{-1}$ beneath the Strait of Juan de Fuca, where underthrust sedimentary rocks of the Olympic Peninsula are present[39], to ~7.0 km s$^{-1}$ near the POLARIS profile on Vancouver Island (Fig. 3). Even with the uncertainty of 0.25 km s$^{-1}$ indicated by the partial anomaly recovery in the checkerboard tests at this depth (see Supplementary Note 1), 7 km s$^{-1}$ is too high to be consistent with a large amount of metasedimentary rock, and the E reflectors here must lie within a predominantly mafic unit[40]. On both of the presented seismic reflection sections, the teleseismic high

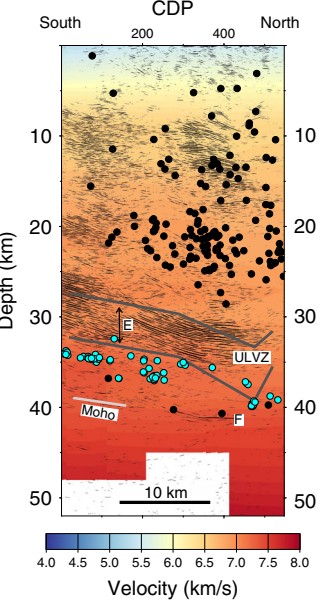

**Fig. 3 Migrated seismic reflection profile on Vancouver Island with superimposed P wave velocity model.** The seismic profile, 84-02, is shown by the brown line in Fig. 1. Low-frequency earthquakes (filled blue circles) occur 0–3 km below the base of the landward dipping E reflectors. P wave velocity increases with depth reaching ~7 km s$^{-1}$ within the E reflections, which are ~6 km thick here. A steeply dipping crustal fault that extends from the surface near CDP (Common Depth Point) 150 to its truncation by the E reflectors at ~30 km depth near CDP 400 has been interpreted from the distribution of crustal seismicity[68]. Filled black circles—earthquakes relocated during 3D tomographic inversion; solid dark grey lines—top and bottom of ultra-low S wave velocity zone (ULVZ) projected from POLARIS (Portable Observatories for Lithospheric Analysis and Research Investigating Seismicity) profile; solid light grey line—Moho of the oceanic plate where it is constrained by wide-angle P wave reflections in a previous 3D tomography study[64]. There is no vertical exaggeration.

Poisson's ratio ULVZ projects onto the deepest 3–5 km of the E reflectors, consistent with the thickness of the S wave LVZ inferred from single-station receiver functions on Vancouver Island[41].

**Models of the inter-plate boundary.** Based on the fact that the LFEs reveal the location of low-angle thrusting during slow slip, we propose two alternative models of the plate boundary in the region of slow slip that are consistent with the structural constraints from seismology:

(1) The inter-plate boundary is thin, perhaps <1 km thick, and corresponds closely to the location of the LFEs where all the slip between the two plates is accommodated (Fig. 5a). In this case, the 3–5 km thick zone of high Poisson's ratio, i.e. the ULVZ inferred from the analysis of teleseismic travel times, must be in the overriding North American plate, and forms the lower part of a thicker inactive shear zone.

(2) Since recorded LFEs account for <1% of the total slow slip[34], slip could be distributed vertically through a thicker inter-plate boundary zone, which may correspond to the 3–5 km thick region of anomalous elastic parameters in the ULVZ immediately above the LFEs (Fig. 5b). Rocks in this zone could be derived from the lower forearc crust or from the upper part of the descending igneous oceanic crust, or perhaps both regions. If upper oceanic crust occurs within the ULVZ, it would be actively deforming, and depending on the vertical distribution of slip,

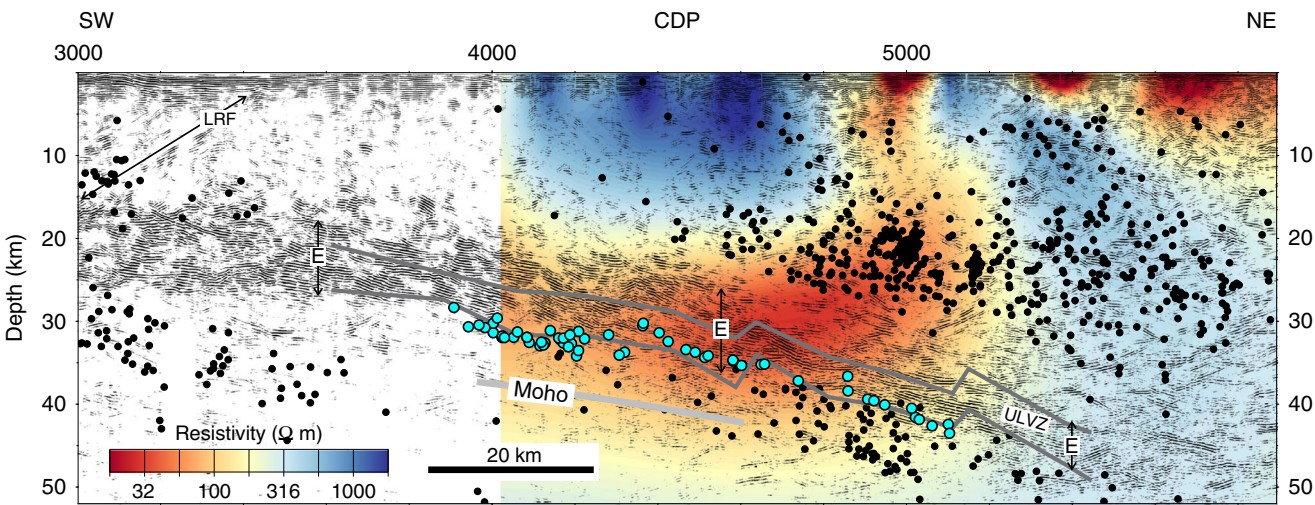

**Fig. 4 Composite migrated seismic reflection profile offshore Vancouver Island with superimposed resistivity model.** Original seismic profiles are shown by the brown line in Fig. 1, but the distance along the profile is incremented evenly along an azimuth of 60°. Low-frequency earthquakes (filled blue circles) occur at the base of the landward dipping E reflectors, which are interpreted to be a 6–10 km thick shear zone, because they truncate overlying faults. A landward dipping zone of relatively low resistivity (30–80 Ω m), which has been projected onto the seismic reflection profile from the inverted 2D section, corresponds approximately to the E reflectors. Filled black circles—earthquakes relocated during 3D tomographic inversion; solid dark grey lines— top and bottom of ultra-low S wave velocity zone (ULVZ) projected from POLARIS (Portable Observatories for Lithospheric Analysis and Research Investigating Seismicity) profile; solid light grey line—Moho of the oceanic plate where it is constrained by wide-angle P wave reflections in a previous 3D tomography study[64]; LRF—Leech River fault at the northwestern edge of the Crescent terrane. CDP Common Depth Point. There is no vertical exaggeration.

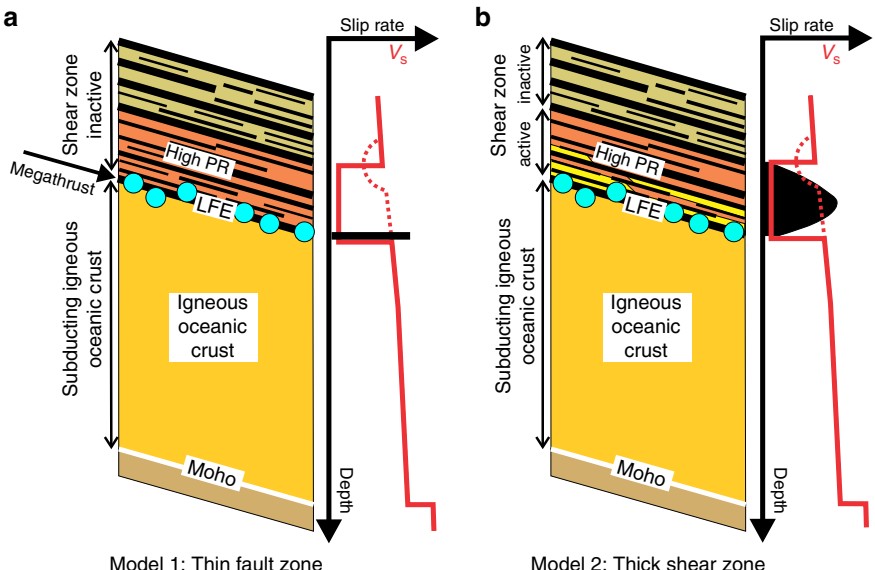

**Fig. 5 Alternative models of the inter-plate boundary beneath southern Vancouver Island in the region of slow slip. a** Model 1: All slow slip on the inter-plate boundary coincides with the low-frequency earthquakes (LFE), implying that the plate boundary is relatively thin and that the overlying E reflectors represent an inactive shear zone. The region of high Poisson's ratio (PR) lies within the overriding plate. **b** Model 2: Slow slip is vertically distributed within the lower part of the reflective shear zone, likely corresponding to the region where Poisson's ratio is elevated. The shear zone includes underthrust rocks from the overriding forearc (orange), which may be imbricated with basaltic rocks derived from the top of the descending oceanic crust (yellow). Seismic velocities beneath southern Vancouver Island are too large to be consistent with a large amount of subducted metasedimentary rock, and the lithology in the shear zone here is largely mafic. Schematic vertical variations in S wave velocity inferred from teleseismic data and tomography are shown by the solid red and dashed red lines respectively; the latter is not correlated with the high Poisson's ratio ultra-low S wave velocity zone (Supplementary Fig. 1d).

parts of the uppermost igneous oceanic crust could be either underplated to the overriding plate or carried down into the mantle or both[36,42]. This zone of active deformation underlies the upper E reflectors which likely arise beneath Vancouver Island in previously sheared and underplated mafic rocks.

Exhumed examples of basalt stripped from the downgoing oceanic crust at depths >20 km are commonly <300–500 m thick[43], leading to the suggestion that the plate interface is typically <300 m thick[42] when the sporadic accretion of thicker sections of oceanic crust is not occurring[44]. In this situation,

which corresponds to Model 1, the LFEs would arise from brittle failure on discrete faults within a relatively narrow zone of deformed rock. If strain is distributed across a broader zone, as in Model 2, then this coupled zone, which could reach 10 km thick, can extend into the upper plate and also cut down into the oceanic slab, perhaps to the top of the sheeted dikes[42]. Strain will occur by creep on multiple faults or shear zones separating stronger or more fluid-rich layers[14,42]. In this case, given their location at the base of the reflective shear zone, the LFEs would occur on the basal decollement of the thick zone of inter-plate coupling.

The LFEs project downdip into the top of a ~10 km thick band of seismicity that occurs in the subducting slab below 40 km depth (Fig. 2 and Supplementary Fig. 2). Farther downdip, this relationship locates a thin plate boundary at the top of the slab seismicity in the case of Model 1, or a thick inter-plate boundary zone immediately above the seismicity in the case of Model 2. In the SHIPS survey through the Strait of Juan de Fuca, wide-angle reflections were recorded from an interface ~7 km below the LFEs (Fig. 4), and this surface is interpreted to be the Moho of the subducting oceanic plate[45]. Downdip projection of the Moho indicates that relocated inslab earthquakes at >40 km depth occur within both the oceanic crust and the uppermost oceanic mantle. A zone of low Poisson's ratio (0.22–0.25 in Supplementary Fig. 1c) and low $(\lambda-\mu)/\rho$ ($<-1.5$ km$^2$ s$^{-2}$ in Fig. 2c) at 53 km depth is associated with this deeper seismicity.

**Resistivity image**. Previous 2D inversions of long-period MT data across southern Vancouver Island (Fig. 1) revealed a landward dipping conductor (low resistivity layer of ~30–80 $\Omega$m)[32,33], comparable to the 30 $\Omega$m conductor inferred beneath central Vancouver Island using 1-D inversion of individual stations and consistent with a porosity of 0.5–3.5% for realistic pore geometry and interconnection[46]. We present new inversions of these MT data, and find that the lowest resistivity of 30 $\Omega$m occurs within a 20–30 km wide region that includes the E reflectors and lies immediately above the onset of the seismicity within the subducting slab at 40 km depth (Fig. 4). The conductor appears to extend seaward in an updip direction with a resistivity of 50–80 $\Omega$m, approximately coincident with the lower E reflectors and the ULVZ; however, the updip depth of the conductor is not well constrained by the inversion because MT stations were only deployed onshore. The conductor could lie deeper, mostly below the E reflectors, with a resistivity of 60–90 $\Omega$m, if a larger trade-off parameter was used in the inversion (see Supplementary Note 2). The thickness of the conductor is also uncertain because the inversion is primarily sensitive to the depth integrated conductivity (conductance). The conductor terminates landward, rising into the overlying crust, above the landward termination of the LFEs, and close to the downdip terminations of the zones of low S wave velocity and high Poisson's ratio inferred from teleseismic data; the location of the conductor's landward termination is well constrained, being a feature of all computed inversions (Supplementary Note 2). Furthermore, if the conductor continued to greater depth, then resolution tests using synthetic inversion of modelled MT data indicate that this longer landward dipping anomaly would be resolved, which is not the case. Modelling also indicates that the fit to the data is improved if the eastern end of the conductor rises into the North American plate (Supplementary Note 2), as indicated by the inversion of the field data in Fig. 4. The 2D resistivity model is consistent with the interpretation that interconnected porosity exists in a landward dipping fluid-rich region close to the western end of the ULVZ, where very high Poisson's ratio and near-lithostatic pore fluid pressures are inferred[22], and that this zone rises into the overlying plate near the conductor's eastern termination.

## Discussion

**Fluids and the inter-plate boundary**. The incoming Juan de Fuca plate contains significantly less water than oceanic plates in other subduction zones, and in northern Cascadia most water is found in the sedimentary section and the upper oceanic crust[47]. In the early stages of subduction, water is released through porosity loss with a relatively minor contribution from low-grade metamorphism, and the remaining water is carried to depth in the form of hydrous minerals. In warm subduction zones such as Cascadia, mineral dehydration reactions in the subducting crust, primarily involving chlorite, amphibole, and epidote are responsible for most water released between 18 and 40 km depth based on thermal-petrologic modelling of subduction zones[48,49]. Beneath the POLARIS profile, and in contrast to the Olympic peninsula to the south[39], it is unlikely that there is a significant volume of subducted sediment near the plate boundary, because the seismic velocities are too high. Thus in the structural interpretation depicted in Model 1 (Fig. 5a), dehydration fluids are primarily generated in the upper igneous oceanic crust below the LFEs, but in the case of Model 2 fluids can also be produced above the LFEs from oceanic crust incorporated into the shear zone (Fig. 5b). The E reflections demonstrate the existence of numerous laterally continuous boundaries within the landward dipping shear zone, with many appearing to extend up to 10 km. These boundaries are likely barriers to vertical fluid migration, trapping fluids whether rising from the subducting plate or produced within the shear zone itself (Fig. 6), and the concentration of fluids here is consistent with the magnitude of the observed electrical conductor[50]. High pore pressures can be maintained through development of low permeability associated with active shear[51,52] within the lower, low-rigidity ULVZ portion of the E-layer, enabling the generation and tidal modulation of slow slip and LFEs near its base[53,54]. In response to spatial and temporal variations in pore pressure, fluids may also migrate updip within the shear zone, consistent with field observations of ductilely deformed quartz veins parallel to foliation in exhumed relict plate boundaries[52,55]. In the upper E-layer, which does not exhibit high Poisson's ratio, high fluid pressures may have previously existed, but have been bled off over time.

The downdip termination of the LFEs at ~45 km depth coincides with the downward limit of the overlying zone of high Poisson's ratio, suggesting a decrease in pore pressure here, which increases the effective normal stress across the fault and changes its mode of failure. The shoaling of the electrical conductor over a distance of ~30 km prior to its landward termination indicates that water is escaping from the shear zone here and rising into the overlying crust, thus lowering the pore pressure within the shear zone. Temporal changes in S wave velocity during slow slip events have been interpreted as evidence of fault-valve behaviour in the same area, i.e. where the LFEs are located at 35–45 km depth[56]. Fluid escape may be related to thinning of the E reflectors from 10 to 6 km as they extend past the corner of the mantle wedge at ~35 km depth, which reduces the thickness of the low permeability barrier, i.e. cap rock, over the high pore pressure zone. In addition, or alternatively, the predicted onset at 40 km depth of eclogitization under equilibrium conditions as appropriate for hydrated metabasalt with its associated volume reduction and strain may disrupt the fluid seal above the LFEs, leading to fluid expulsion from the slab at this level[25,26,56,57] and serpentinization of the overlying mantle wedge[58,59]. An extensive 10–15 km thick region of anomalous mantle, which has a Poisson's ratio of 0.26–0.28, exists between the base of the continental crust and the downgoing

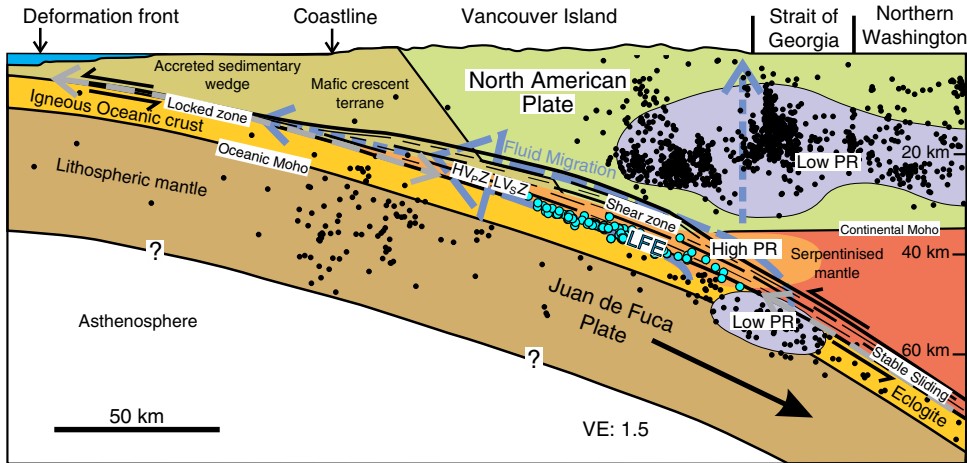

**Fig. 6 Migration and trapping of metamorphic fluids in shear zone above dehydrating Juan de Fuca plate.** Dehydration of the subducting plate, including that due to transformation of basalt to eclogite, is associated with in-slab seismicity at >40 km depth. The onset of seismicity corresponds to a zone of lower Poisson's ratio in the slab, which lies below an extensive region at the tip of the mantle wedge where seismic velocities and elevated Poisson's ratio (PR) are consistent with partially serpentinized peridotite. In the case of plate boundary Model 1, rising fluids are trapped within the shear zone above the subducting plate and may migrate seaward (blue dashed arrows), with local escape possible in regions of higher vertical permeability. In the case of Model 2, fluids are also generated in the shear zone by dehydration of upper oceanic crust stripped from the downgoing slab. In both cases, high pore pressures and high Poisson's ratio occur within the lower part of the shear zone, but pore pressures fall where slab-derived fluids escape through the mantle wedge, landward of the downdip termination of the low-frequency earthquakes (LFEs) and electrical conductor (vertical dashed blue arrow). The shear zone between the mantle wedge and the subducting plate can include fragments of oceanic crust, eroded forearc and/or rocks of the mantle wedge. $HV_PZ$ high P wave velocity zone, $LV_SZ$ low S wave velocity zone, LFEs—filled blue circles, earthquakes—filled black circles.

oceanic slab. With P wave velocities of 7.1–7.7 km s$^{-1}$, the elastic properties in this seaward part of mantle wedge are consistent with peridotite that has been partially serpentinized by 20–60%. Water has likely risen further into the continental crust, and this fluid flux may be responsible for concentrations of seismicity and Poisson's ratios as low as 0.225 that are interpreted as due to fluid-aided metamorphism[25].

**Structural complexity of the plate boundary**. At present in the region of slow slip, the depth of the northern Cascadia inter-plate boundary is only unambiguously constrained by the small amount of slip associated with the LFEs. Though it is likely that the plate boundary also includes the 3–5 km thick zone of anomalous elastic properties in the lower part of the landward dipping reflectors just above the LFEs, where active slip could maintain porosity and high pore pressure, there is currently no well-constrained evidence of slip at this higher level; the few LFEs that appear to occur within the ULVZ could be due to location errors. In the current study, we are unable to resolve whether slip occurs on a single or multiple, vertically stacked surfaces, some of which may lie within the reflective shear zone. The existence of additional fault zones that form part of a more complex plate boundary[60] also cannot be excluded. Discrimination between these various hypotheses likely depends on improved seismic velocity models and better depth location of additional LFEs or the more pervasive non-volcanic tremor[34,61] that will reveal where slip occurs.

## Methods
**3D seismic tomography**. The 3-D P and S wave velocity models were independently obtained by double-difference tomography[62] of airgun shots recorded during the 1998 SHIPS survey and local earthquake data, and have been previously published[25].

Earthquake data included the P and S wave arrival picks of 4725 events between 1992 and 2012 from the catalogue of the Geological Survey of Canada. An additional 333 earthquakes between 2002 and 2006 that were detected automatically using cross-station correlations were also used. P and S first arrivals were manually picked at all permanent and temporary stations in the study area, including the POLARIS array, the three small aperture arrays of the 2004 Deep

Tremor Project, the Plate Boundary Observatory Borehole Seismic Network, the Canadian National Seismograph Network, and the Pacific Northwest Seismic Network. P and S wave arrival picks from events between 1980 and 2002 that were used in an earlier study[63] in northern Washington State were also included.

A catalogue of 276 LFE template correlations from southern Vancouver Island was employed in the tomography. Each template comprises the stacked waveforms of up to a few thousand independent LFE events detected by network cross-correlation that have identifiable P and S wave arrivals. The LFE templates provide valuable additional constraints on the subsurface velocity variation, because they occur in regions of the crust where there is little conventional seismicity. Waveform cross-correlation delays were computed from all 2002-2006 earthquakes and the LFE templates

During the 1998 SHIPS experiment, P wave arrivals from airgun shots in the Strait of Georgia, Strait of Juan de Fuca and Puget Sound were recorded by temporary land stations and ocean-bottom seismometers. The dataset of P arrival picks was subsampled by a factor of three, and included in the tomographic inversion, providing valuable constraint on the upper crust, particularly in areas where there are few earthquakes.

A previously derived 3D P wave velocity model[64], linearly interpolated onto a $12 \times 12 \times 3$ km grid, was used as the starting model for the tomographic inversion of the P arrivals. This model was scaled by 0.5774 to obtain a starting model for inversion of the S wave arrivals. 28 iterations of TomoDD, alternating between relocation of hypocentres and joint inversion for hypocentres and velocity, were run with greater weight placed on differential time measurements in the later iterations to improve spatial resolution. The root mean square travel time residual was reduced from 1.4747 s to 0.1195 s, while the residual for the cross-correlation data improved from 0.2862 s to 0.0442 s. To obtain P and S wave velocity models from which Poisson's ratio could be calculated with a low raypath bias, the tomographic inversion was then run on a travel time dataset restricted to stations at which both P and S wave arrivals were recorded, producing a travel time residual of 0.0924 s. Spatial resolution tests[25] indicate the recovery in the central sections of the model of both a $24 \times 24 \times 6$ km checkerboard with 10 % amplitude perturbation and a landward dipping slab with a $Vp/Vs$ ratio of 2.35 (Poisson's ratio of 0.39), as described in Supplementary Note 1.

The horizontal and vertical errors of the hypocentres input to the tomographic inversion were $+0.7/-0.8$ km and $+1.3/-1.8$ km respectively, and these errors will likely be reduced by the inversion. Explicit post-inversion uncertainties are unavailable using this tomographic method, though the mean centroid shift of the hypocentres was 0.37 km. We conservatively consider the typical horizontal and vertical errors in relocated hypocentres to be $+/-1$ km and $+/-2$ km respectively.

**Seismic reflection section**. The seismic section displayed in Fig. 3 is the coherency-enhanced migration of line 84-02 available from the Geological Survey of Canada. The composite seismic reflection image in Fig. 4 was constructed with an equal trace increment every 50 m along an azimuth of 60° from the unmigrated

stacks of five lines: 85-05 shot by Geological Survey of Canada plus JDF-1, JDF-5, JDF-3, and SG-1 shot as part of the SHIPS program. Prestack processing included bandpass filtering, automatic gain control with a 0.5 s window, muting, normal moveout and stack. The composite stack section, after attenuation of steeply dipping coherent noise, and migrations are shown in Supplementary Note 3 to illustrate the original stack data quality and the construction of the migrated images.

**2D magnetotelluric inversion**. The magnetotelluric (MT) data used in this study are a subset of data acquired for a larger study of the southern Canadian Cordillera whose characteristics have been previously described[32]. A total of 23 long-period MT stations were selected on a profile that extended from the Pacific Ocean to the volcanic arc. A strike direction of N45°W was used for the 2D inversion, consistent with both prior studies of these MT data[32,33].

The MT data were inverted for a 2D resistivity model using a nonlinear conjugate gradient algorithm[65]. The inversion focussed on the transverse magnetic (TM) mode and the vertical magnetic field transfer functions. TM phases exceeding 90°, were excluded from the inversion, because these out-of-quadrant phases cannot be fit by the 2-D inversion; these anomalous phases are most likely caused by channeling of electric currents by localized conductors[66] For all other MT data, an error floor of 20% in apparent resistivity and 5% in phase was applied. Note that a 5% uncertainty in apparent resistivity corresponds to an uncertainty of 1.45° in the phase. An error floor of 0.06 was applied to the vertical magnetic field transfer function. Data uncertainties that were below these floor values were set to the error floor. By choosing a larger error floor for the apparent resistivity, less emphasis is put on fitting the apparent resistivity data, as these might be affected by galvanic distortion.

The inversion began from a 100 Ωm half-space model that included a 0.3 Ωm layer to represent the Pacific Ocean. The inversion algorithm solves for the smoothest 2-D resistivity model consistent with the measured MT data, but the constraint on spatial smoothness can be relaxed at known interfaces. In this study a discontinuity was permitted at the base of the Pacific Ocean and accretionary prism, which prevented the low resistivity ocean layer being smoothed to unrealistic depths. The inversion was not permitted to solve for static shift coefficients as this led to coefficient values that were all less than unity on Vancouver Island, resulting in the dipping conductor beneath Vancouver Island being very shallow.

Geophysical inversion is an inherently non-unique process and the inversion models were assessed in two ways. Firstly, the regularization parameter was investigated. This parameter τ controls the trade-off between the competing requirements of fitting the MT data and finding a model that is spatially smooth. A preferred value of $\tau = 5$ was selected that resulted in a final misfit of 1.66. The second stage used a synthetic MT inversion study to evaluate model resolution. These two tests are described in more detail in the Supplementary Note 2.

**Depth comparisons**. The depths of features interpreted from the teleseismic data were originally calculated from a 1D velocity model used by the Geological Survey of Canada (GSC) for the location of local earthquakes. To locate these features within the 3D tomographic velocity models, a correction was applied for the difference between the 1D and 3D models. The depths of the LVZ, and another feature at 25–35 depth (dashed grey lines in Fig. 2d), and the ULVZ inferred from a relative travel time analysis of Ps and PpS phases (solid dark grey lines in Fig. 2d) were converted to time along vertical paths using the 1-D S wave velocity model, and then back to depth using the 3-D S velocity model derived by double-difference tomography. In practice this correction made little difference in the depths of the landward dipping features, but the low S velocity anomaly at 205–255 km distance in Fig. 2 is ~2 km shallower owing to the faster crustal velocities in the GSC model than determined by 3D tomography in the continental crust. A similar approach was taken to correct the reflection points from the oceanic Moho that were originally located in depth by 3-D P wave tomography[63], resulting in depths in our model that are ~1 km shallower (Fig. 4). These changes in depth provide a characterization of the uncertainties in the absolute depth (from mean sea level), but since all final depths in this paper were obtained using the same TomoDD 3D velocity model after conversion of originally interpreted structures to time, the relative depth uncertainties are likely to be less.

## Data availability
Seismic tomography: Input P and S travel times, recording station coordinates, output velocity models and relocated events are available from the PhD thesis archive of the University of British Columbia (https://open.library.ubc.ca/cIRcle/collections/ubctheses/24/items/1.0371609). Seismic reflection images: Semblance enhanced migration and coordinates of LITHOPROBE line 84-02 are available in SEGY format from the archive of the Geological Survey of Canada (https://open.canada.ca/data/en/dataset/f96393c4-29c7-5624-80cd-046a1496b4c0). Stack, migration and coordinates of line 85-05 available in SEGY format from the archive of the Geological Survey of Canada (https://open.canada.ca/data/en/dataset/32bbd280-b99a-5b07-af55-71fcbdb95e16). Stack, migration, and location of composite seismic section around southern Vancouver Island are available from the authors in SEGY format on reasonable request. Raw shot gathers and coordinates for the SHIPS seismic lines are available in SEGY format from the

Incorporated Research Institutions for Seismology (http://ds.iris.edu/ds/nodes/dmc/data/types/waveform-data/). Magnetotelluric data: Input data and output resistivity models are available from the authors in EDI format on reasonable request. Digital elevation model: Topography displayed in Fig. 1 was obtained from ref. [67]. (https://pubs.usgs.gov/of/1999/0369/).

## Code availability
Seismic tomography: TomoDD code can be obtained from the original authors[62]. Seismic reflection processing: ProMAX software is available from Halliburton Corp under a commerical licensing arrangement; additional module to compute segment migration is available from the authors on reasonable request. 2D magnetotelluric inversion: WinGLink software is available from Schlumberger Corp, Milan, under a commerical licensing arrangement. Projections: Software to project earthquakes, LFEs, and resistivity values onto random line through 3D velocity model is available from the authors on reasonable request. Map: Fig. 1 was created using GMT (Generic Mapping Tools), which is available from https://www.generic-mapping-tools.org/.

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

## Acknowledgements

Tom Brocher provided the travel time picks from the SHIPS experiment and some earthquakes in Washington State that were used by GS in the tomographic inversion. Simon Peacock provided helpful advice on characterization of the metamorphism. This project was supported by the Natural Sciences and Engineering Council of Canada.

## Author contributions

A.C. and M.B. interpreted the seismic data and wrote the paper; G.S. ran the 3D tomographic inversions and interpreted the seismic velocity models; MU inverted the magnetotelluric data and interpreted the resistivity model.

## Competing interests

The authors declare no competing interests.

**Additional information**

