## [Peer Review File · Nature Communications]

Reviewers' comments:

Reviewer #1 (Remarks to the Author):

I completed my review of the manuscript entitled "The slow-slip inter-plate boundary in the northern Cascadia subduction zone" by Dr. Calvert and co-authors (manuscript number NCOMMS-20-05972-T), which has been submitted for publication in Nature Communications. I recommend that the work by Calvert et al. is published in Nature Communications after a revision because (1) it is technically sound, (2) it provides strong evidence for its conclusions, and (3) it is important to scientists in the field of subduction zone studies. Regarding (4), the general criteria related to novelty, I do not find the presented work highly novel but, overall, it does represent an advance in our understanding that will influence thinking in the field. More specifically, this work brings the various lines of evidence and thought on the location of the subduction interface and the related deformation in what to me appears to be the most logical direction. The submitted Calvert et al. work is impactful, wonderfully written, and a real joy to read but can be improved by suggested modifications and additional credit to the earlier work that has pointed in the same direction.

Points to address as requested by Nature Communications:

1. What are the major claims of the paper? I think that the major claim of this paper is that it can convincingly reconcile the various earlier interpretations on the location of the subduction interface and the related deformation. Some of these earlier interpretations, which were often based on different types of datasets, differ in a major way, so the work accomplished by Calvert et al. was a challenging task.

2. Are the claims novel? If not, please identify the major papers that compromise novelty. Much of what is claimed is novel but not everything. Nedimović et al. in a 2003 Nature paper, interpret the location of the subduction interface to be at the exact same location as Calvert et al. in this work, and directly point to the tie between the change in the width of megathrust reflectivity and the change in the deformation character at the megathrust, from narrowly focused brittle deformation at shallower megathrust depths to broadly (vertically) distributed deformation at greater megathrust depths. In even earlier papers authors have pointed to various interpretational possibilities for the thick band of megathrust reflectivity but not to a change from a narrow trust with focused deformation to vertically distributed deformation at greater depth. Nedimović et al. were fortunate to carry out and publish their work soon after the discovery of the episodic tremor and slip, which resulted in better understanding of the change in the deformation style on the megathrust as a function of depth. The Nedimović et al. hypothesis on the relationship between the megathrust reflectivity width and megathrust vertical deformation distribution led to large funding from NSF to test this hypothesis at other places such as the Alaska-Aleutian subduction zone and the rest of the Cascadia subduction zone. The 2011 ALEUT experiment across a large chunk of the Alaska-Aleutian subduction zone, which unlike Cascadia is a cold subduction environment, recorded the exact same reflection signature on the megathrust as previously observed at the northern Cascadia subduction zone, one that changes from a very narrow reflection package (hundreds of meters) at shallow depths to a thick one (some 6 km) at greater depths. These results were published in 2015 by Li et al. in JGR Solid Earth and also include modeling to prove the megathrust origin of the thick reflection band and a schematic showing how the coupling changes in terms of asperity size and distribution (horizontally and vertically) when the deformation focus changes from narrow to broad vertical distribution. I don't see how acknowledging these earlier achievements can have a negative effect on publishability of the submitted Calvert et al. manuscript in Nature Communications because the authors present a convincing interpretation from multiple lines of evidence for the location of the megathrust and also narrow the focus of the vertically distributed deformation to the bottom half of the thick reflection zone. These are important new contributions. Moreover, if the authors disagree about the level of acknowledgement I believe the earlier work deserves, this is all right with me, as I don't want my comments to stand in the way of the important results presented in the submitted manuscript.

3. Will the paper be of interest to others in the field? Absolutely! I think that this paper will serve as a guide to other researchers on how to interpret various data sets and results collected and produced at subduction zones and how to integrate them into a consistent interpretation for the location of the subduction interface and the related deformation.
4. Will the paper influence thinking in the field? Please see above.
5. Are the claims convincing? If not, what further evidence is needed? The presented interpretation is quite convincing and no other lines of evidence are needed. This is an integration paper.
6. Are there other experiments that would strengthen the paper further? How much would they improve it, and how difficult are they likely to be? I don't think that there is a need for any additional new information for this paper.
7. Are the claims appropriately discussed in the context of previous literature? For the most part yes, but not fully for the previous multi-channel seismic reflection work. Please see explanations in comment 2.
8. Is the manuscript clearly written? If not, how could it be made more accessible? Yes, the manuscript is written clearly and is a pleasure to read.
9. Could the manuscript be shortened to aid communication of the most important findings? Not taking into account the rules for manuscript length set by the journal, because I did not look into such details, I suggest no shortening for the manuscript. However, if the manuscript is considered too long by the journal, I would suggest cutting down on some parts of the methodology because most of the results were produced as part of earlier publications and do not need to be repeated.
10. Have the authors done themselves justice without overselling their claims? It appears to me that the manuscript mildly overstates the novelty of the interpretations carried out. But this is easily fixable. Please see comment 2.
11. Have they been fair in their treatment of previous literature? Please see comment 2 and further text with specific comments to address.
12. Have they provided sufficient methodological detail that the experiments could be reproduced? My feeling is that there is perhaps more than enough detail but clarity of presented information can be improved. This particularly because it is hard to know what are the new results the authors produced for this paper and what was done for previous publications. Clarifications are needed on this in the revised manuscript. For example, was the LFE relocation and related 3D tomography in the Methods section done exclusively for this manuscript or was it done and described in an earlier another publication, like for most of the other seismological results presented. There also is a lot of information (a whole section) on the resistivity work, all of which seems published in a number of earlier papers. I don't think that any of that takes away from the significance of the work the authors have carried out, but I would like to see an explicit description of which results are new and done for this work and which are not and are used and described in earlier work and publications.

Other points to address as requested by this reviewer:

1. I would like to see the Abstract rewritten to focus more on what is new in this manuscript. As is, the Abstract starts with summary description of previous results/interpretations, and then moves to new results done for this manuscript, which are LFE relocations and a new 3D velocity model. This is followed by the classic "Here we show....", all as one would expect. However, the Abstract then veers off the expected path as it describes magnetotelluric results, which I think we know from earlier papers. To me what is new in this manuscript is simple, its about (1) where the

various sources of data and results point in terms of the location of the megathrust and (2) where the same data point to in terms of the deformation distribution, from focused at shallower depths to distributed at greater depth but now narrowed down to about the lower half (~3 km) of the thick reflection band. I think that being more direct and bold with their interpretation results instead of leaving the reader to work hard to figure out what the important finds are would make a much better Abstract. As is, the Abstract includes even numbers and units, such as the 30-80 Ohm-meter electrical conductor. This does not tell the reader much as one does not know how does that compare to layers above and below in terms of numbers. Nevertheless, I have no objections to keeping the Abstract as is but I do think that the current one is not the most effective at conveying the key important finds of their work but is instead convoluted.

2. Lines 12-16 and 52-56. The manuscript explains at various places, starting from the Abstract, that the oceanic crust is considered to be overpressured in many interpretations owing to fluid trapped beneath an impermeable seal along the overlying inter-plate boundary. This agreement among what appear to be many interpretations is, perhaps, not unexpected considering that, best to my knowledge, none of these interpretations actually study and consider the hydration of the incoming oceanic crust/plate but rather take into account some generic model of oceanic crust/plate. I don't think that such studies are effective because the amount of water in pores and chemically bound in the subducting oceanic lithosphere varies by orders of magnitude from one subduction zone to another. Cascadia subduction zone has the globally best constraints on the water content of the incoming oceanic lithosphere that are provided in a recent Nature Geoscience paper by Canales et al. (2017). This paper constrains the amount of pore and chemically bound water layer by layer (sediments, layer 2A, layer 2B, Layer 3 and uppermost mantle). We know that the Juan de Fuca is at the extreme dry end of subducting plates. There is a large amount of text in the Calvert et al. manuscript that explains the results from previous work. Some of it is needed because this is an integration paper. However, I fear that integrating all these results about deep and complex subduction structures that are known to have significant uncertainty and low resolution (note that there basically are no uncertainties given in the manuscript, nor any resolution tests) will not give the reader high confidence in the presented interpretation, which I actually think is well done. Why not integrate the detailed results on the hydration of the subducting Juan de Fuca plate and tell the reader where the water is coming from (which lithospheric layer) and how much water one could expect to have released and where, and how does that agree with what is observed in the resistivity and other results? The authors have a real opportunity to bring subduction zone interpretation to a new level and it would be a shame to miss this opportunity. Of course, the authors can argue that this is out of the scope of their paper. However, at minimum, they need to put their interpretations in the context of the existing Juan de Fuca plate hydration knowledge and there is a lot of work done and published in the last 10 years.

3. Lines 48-56 and 137-143, etc. Looking at the ideas and references presented in this manuscript, as well as from my reading of other earlier papers, I note that the various interpretations on the role of the subducting oceanic crust and lithosphere appear to lack in-depth information on oceanic crustal structure. These interpretations are challenging for me to understand as they lump the oceanic crust into upper and lower and then treat the upper crust as a homogeneous body. However, the upper crust has two layers, A and B, that are vastly different in terms of rock type, structure, porosity, anisotropy, alteration minerals filling in pore space, mechanical properties, etc. Layer 2A is made of basaltic extrusive rocks that can be represented by subhorizontally layered flows that were recently beautifully imaged (see Arnulf et al., 2014 Geology paper). Layer 2A can have a structure that has subhorizontal surfaces of mechanical weakness that could shear after subduction at great depth. However, layer 2B, which represents say 80-90% of the upper crust, is composed of vertical diabase dykes and, as such, should show greatest mechanical resistance to shearing in horizontal or subhorizontal direction of all the oceanic or continental lithospheric layers. I just simply cannot see a mechanism that would make it form horizontal or subhorizontal shear zones that exhibit reflectivity in that direction. Here I am discussing the subduction depths/pressures/temperatures targeted in the submitted manuscript, which seem to be too shallow/small/low for significant metamorphic processes in the oceanic lithosphere to take place. So, in summary, including layer 2A into the imaged and interpreted

shear zone is acceptable from my perspective but including layer 2B is not. However, including layer 2A adds only a few hundred meters to the shear zone and cannot be significant for formation of a shear zone that is 6 km or so thick. I suggest removing or modifying Figure 4c interpretation. If the authors still want to include the upper oceanic crust into the horizontal/subhorizontal shear zone that represents the plate interface area exhibiting distributed deformation, then they need to explain both (1) why would the oceanic crust in general start shearing horizontally/subhorizontally before the lower continental crust, especially considering that it is the lower continental crust that should be much more quartz rich, and (2) how can layer 2B, which is made of vertical diabase dykes, a mechanically strong material with vertical surfaces of preferential weakness, shear horizontally/subhorizontally. Therefore, it appears to me that the submitted manuscript requires some reevaluation of the earlier work on receiver function interpretation, all of which focuses on the LVZ as being due to part of the oceanic crust.

4. Lines 63-64. "LFEs lie at the base of a regionally extensive shear zone". Does this mean below it or within its lowermost part? Would be good to be specific because when reading this it seemed to me inconsistent with what is written in the Abstract on lines 13-16. Please be exact throughout the text. At the same time, this could be just me picking up on something that is not necessarily a problem.

5. Lines 118-122 and elsewhere. In his research, Calvert et al. are relying on crustal reflection sections that are extremely coherence filtered, more than I have seen anywhere else except in their earlier work. The extreme coherency filtering makes these reflection sections look artificial to me and make me question the interpretations derived from them. Could the authors provide in supplementary materials or in their rebuttal a comparison supplementary figure that shows the reflection section from Figure 3 together with its equivalent that is not coherency filtered and moderately coherency filtered so that we can see if the ~6-10 km thickness of the wide reflection band is observable in these sections as well. The 10 km thickness and lateral extent of >10 km (as stated in line 182) seems to me to be possibly too large.

6. Lines 133-135. Interpretations are made here but also elsewhere in the manuscript based on the values of various properties derived from the collected data such as P and S wave velocities, change in these velocities from the starting model, resistivity, etc. But nowhere in the manuscript I did see any information on the uncertainty these numbers have. My guess is that the resolution is low and uncertainties high for the ~7 km/s velocity mentioned here for the E reflections area discussed. Would it be possible to give error bounds for the ~7 km/s velocity and any resolution tests for these velocity models? This would make the conclusions that these are not metasediments, which I agree, stronger.

In summary, this manuscript comes from an excellent and productive team of researchers. Their presented work is very interesting and deserves to be published in Nature Communications after a revision.

Mladen Nedimović

Reviewer #2 (Remarks to the Author):

The manuscript presents new earthquake relocation results in the context of multiple structural seismology images and a magnetotelluric image of the northern Cascadia subduction zone. The new results, particularly the low-frequency earthquake hypocenters, are used to motivate a new synthesis of plate boundary structure from imaging and earthquake studies focused on a transect beneath southern Vancouver Island. While there is already a sequence of papers making inferences based upon some of the results presented here, the text is right to point out that there still is not agreement on the location of the plate interface with respect to the commonly imaged low-velocity zones, broadband earthquakes, and low-frequency earthquakes. This manuscript makes a compelling attempt to weave together all these lines of evidence with consistent modeling of P

and S velocity structure, reflection/conversion imaging, earthquake locations, and conductivity. The manuscript's integrative results and interpretation rise to the level of a significant new contribution despite the extent of prior research on the subduction interface in this region.

The manuscript is clearly written and the supplemental description of method is helpful. Hypothesis-driven resolution tests seem to be the only potential ingredient that is missing. It would be helpful to know more about what level of resolution of 3D hypocenters and LVZ (and conductivity) boundaries is feasible given the data used here (source-receiver geometry, frequency content, etc.). I realize this is not a small task given the breadth of data types considered here, but it would be valuable to have greater treatment of the resolution limits in this manuscript rather than just references to other studies. It would strengthen the interpretations if synthetic data for a hypothetical model were inverted for seismic structure & hypocenters and conductivity structure.

Reviewer #3 (Remarks to the Author):

Review Calvert et al.

General comments

I am not over familiar with the literature for this margin, and this is probably a central reason for me finding this paper a bit difficult to read. So from my perspective, the reader needs a bit more help to: understand the main goals and principal findings of the study, distinguish between what's already known (previous geophysical results and models) and what's new in the manuscript. The aims and objectives of the current study are not specifically stated anywhere. I agree that these can be partially inferred from the current text, but the paper will have much more impact if the story is better told. On the first read of the paper, I initially thought that the P and S wave tomographic models, relocation of the LFEs, and MT results were all new, but the citing in lines 60-62 and 82-83 suggests they have already been published in reference 20, 8, 26 and 27. And when I looked at the figures, I wondered whether the aim was to test between the different models shown in Fig. 4, but there doesn't seem to be any conclusion about which of these best explains the available data. Then I wondered whether Figure 5 was the principal result – which it might be – but this figure isn't actually cited in the text.

Although Nature Communications papers are formatted slightly differently, they do recommend having a 'Results' section. Papers (the few that I skimmed) do seem to also have a classic introduction, where the 'state of play' is described just before the results section, and there is a clear statement of the aims and objectives of the current study, how these aims are achieved, and what results have been achieved. So I think the paper could be improved with some re-structuring, with clearer statements about what is known and what is new, include a statement of the aims and objectives of the study, and have a separate results section. Perhaps the aim is to distinguish between the three alternative models of the boundary (fig. 4), or better locate the geophysical properties across the inter-plate boundary, and the approach is to use existing data and generate some better geophysical plots (i.e. figures 2 and 3)?

In summary, as currently written, it is not clear to me whether there is sufficient scientific advancement in the paper to merit publication in Nature Communications.

Detailed comments

In lines 27-68 and 170-223, the text jumps about a bit too much with not enough linking between concepts, contains sentences that appear to be a distraction from the main story, and the text is a bit too passive (over usage of 'can' and 'may').

In lines 66-68 "Consequently, features previously associated with the subducting oceanic plate

either lie within the overriding plate or form part of a plate boundary zone a few km thick", it is not clear exactly what features are being referred to or why it is important. The linking between statements like this one and the text below is poor – which makes the story difficult for the reader to follow.

Line 89 – is the identification of the ULVZ new, or is it already published in reference 17, or is reference 17 a citation for the adopted methodology.

Line 106 Can tomographic inversions make associations?

Line 121 – I am going to assume that I have permission to moan here. The interpretation that the E reflectors are formed by shearing is an interpretation – not one that I personally think is correct. The E reflections look like normal layered lower continental crust, into which faults in the upper continental crustal faults merge, as expected for a brittle to ductile transition. We don't know what causes lower crustal reflectivity (there is no consensus), but it is much easier to generate high-amplitude reflections with lithological differences, such as intrusion of mafic igneous rocks into the lower continental crust (which would be consistent with an average velocity of 7 km/s). Lithological layering (as the cause of the reflectivity) does not preclude the occurrence of shearing after the formation of the reflectors. And intrusions would be easier to envisage as (semi) impermeable barriers. Is there a good reason to exclude this? Anyway, I'd like to request that the revised text is more equivocal about the cause of the reflectivity.

The scenarios shown in Figure 4 are first noted in lines 124 and 127, and then later they are referred to as model 1 and model 2 (lines 144-147). It would help the reader if there was a clear statement (on their first mention) that there are three models that may explain the data, and for the features in each of the proposed models to be explained and discussed. Is there a preferred model? This is not clear from lines 179-180.

Resistivity model (image suggests something that is well-resolved). MT data have a relatively low resolution and we can expect recovered resistivity anomalies to be a very smooth version of reality with not very good anomaly thickness and depth control. With this in mind, the use of the words "consistent with" is fine, but stronger assertions should be avoided.

The red zone looks to be between CDPs 4400 and 5100, be relatively flat or slightly curved at its base, and the top of the red zone is curved upwards towards the surface in the landward direction. Is there a good reason to be confident about the dip and depth of the yellow/orange resistivities, to be able to say it is approximately coincident with the ULVZ?

Line 159, the resistivity and E reflectors don't correlate that well. The red zone correlates with the E reflectors at about CDP 4600, but at higher CDPs the red zone is above the E reflectors.

Line 166 Sentence "Broadly consistent with" needs re-writing. Consistent with the interpretation that there is a...

Line 185-186. This statement is too strong – the electrical conductor is not coincident with the E reflectors.

Figure 5. This figure isn't cited.

Prof Joanna Morgan, Imperial College London

Response to Reviewers

We sincerely appreciate the time and effort that the reviewers and Associate Editor have invested in handling our paper, especially in these challenging times. We list **in red** our response to the reviewers' comments immediately following the paragraphs to which they apply, and provide the introductory comment below:

Our paper significantly impacts two key debates:

- 1) **We refine current interpretations of the inter-plate boundary to the lower part of the previously mapped shear zone, and show that the recently discovered, enigmatic low frequency earthquakes are distributed along its base, supporting their interpretation as shear ruptures.**
- 2) **We show that in Cascadia the low velocity zone identified by teleseismic receiver functions, which has been previously interpreted to be subducting oceanic crust, actually lies in the overriding plate or arises from the inter-plate boundary. So inferences of metamorphism in the subducting crust from the landward termination of the low velocity zone should be reconsidered.**

We further make the secondary point that much of the contradiction between the interpretations of different seismic results can be traced to the use of different seismic velocity models.

Reviewer #1 (Remarks to the Author):

I completed my review of the manuscript entitled "The slow-slip inter-plate boundary in the northern Cascadia subduction zone" by Dr. Calvert and co-authors (manuscript number NCOMMS-20-05972-T), which has been submitted for publication in Nature Communications. I recommend that the work by Calvert et al. is published in Nature Communications after a revision because (1) it is technically sound, (2) it provides strong evidence for its conclusions, and (3) it is important to scientists in the field of subduction zone studies. Regarding (4), the general criteria related to novelty, I do not find the presented work highly novel but, overall, it does represent an advance in our understanding that will influence thinking in the field. More specifically, this work brings the various lines of evidence and thought on the location of the subduction interface and the related deformation in what to me appears to be the most logical direction. The submitted Calvert et al. work is impactful, wonderfully written, and a real joy to read but can be improved by suggested modifications and additional credit to the earlier work that has pointed in the same direction.

Please see introductory comments above.

Points to address as requested by Nature Communications:

1. What are the major claims of the paper? I think that the major claim of this paper is that it can convincingly reconcile the various earlier interpretations on the location of the subduction interface and the related deformation. Some of these earlier interpretations, which were often based on different types of datasets, differ in a major way, so the work accomplished by Calvert et al. was a challenging task.

In our longer introduction, we have made clearer the nature of the discrepancy between the results from active source and teleseismic methods, and explained how our study addresses this problem in northern Cascadia.

2. Are the claims novel? If not, please identify the major papers that compromise novelty. Much of what is claimed is novel but not everything. Nedimović et al. in a 2003 Nature paper, interpret the location of

the subduction interface to be at the exact same location as Calvert et al. in this work, and directly point to the tie between the change in the width of megathrust reflectivity and the change in the deformation character at the megathrust, from narrowly focused brittle deformation at shallower megathrust depths to broadly (vertically) distributed deformation at greater megathrust depths. In even earlier papers authors have pointed to various interpretational possibilities for the thick band of megathrust reflectivity but not to a change from a narrow thrust with focused deformation to vertically distributed deformation at greater depth. Nedimović et al. were fortunate to carry out and publish their work soon after the discovery of the episodic tremor and slip, which resulted in better understanding of the change in the deformation style on the megathrust as a function of depth. The Nedimović et al. hypothesis on the relationship between the megathrust reflectivity width and megathrust vertical deformation distribution led to large funding from NSF to test this hypothesis at other places such as the Alaska-Aleutian subduction zone and the rest of the Cascadia subduction zone. The 2011 ALEUT experiment across a large chunk of the Alaska-Aleutian subduction zone, which unlike Cascadia is a cold subduction environment, recorded the exact same reflection signature on the megathrust as previously observed at the northern Cascadia subduction zone, one that changes from a very narrow reflection package (hundreds of meters) at shallow depths to a thick one (some 6 km) at greater depths. These results were published in 2015 by Li et al. in JGR Solid Earth and also include modeling to prove the megathrust origin of the thick reflection band and a schematic showing how the coupling changes in terms of asperity size and distribution (horizontally and vertically) when the deformation focus changes from narrow to broad vertical distribution. I don't see how acknowledging these earlier achievements can have a negative effect on publishability of the submitted Calvert et al. manuscript in Nature Communications because the authors present a convincing interpretation from multiple lines of evidence for the location of the megathrust and also narrow the focus of the vertically distributed deformation to the bottom half of the thick reflection zone. These are important new contributions. Moreover, if the authors disagree about the level of acknowledgement I believe the earlier work deserves, this is all right with me, as I don't want my comments to stand in the way of the important results presented in the submitted manuscript.

We recognise that our initial manuscript should have included more discussion and citation of earlier work, and we have addressed this point in the introduction by discussing the papers by Nedimovic et al. (2003) and Li et al (2015), and noting the importance of the landward thickening of the seismic reflectors. We also point out that in the absence of well-located low-angle thrust earthquakes it is difficult to unequivocally associate these reflections with the inter-plate boundary, and that some plate boundary interpretations, e.g. Sumatra, appear to require a non-reflective fault zone. Hence the need to improve the integration of different datasets, and ideally to use the same velocity models (both P and S wave) in their analysis. We have also added a later paragraph in the Results section to relate our proposed models to previous geological work on exhumed subduction plate boundaries.

3. Will the paper be of interest to others in the field? Absolutely! I think that this paper will serve as a guide to other researchers on how to interpret various data sets and results collected and produced at subduction zones and how to integrate them into a consistent interpretation for the location of the subduction interface and the related deformation.

4. Will the paper influence thinking in the field? Please see above.

5. Are the claims convincing? If not, what further evidence is needed? The presented interpretation is quite convincing and no other lines of evidence are needed. This is an integration paper.

6. Are there other experiments that would strengthen the paper further? How much would they improve it, and how difficult are they likely to be? I don't think that there is a need for any additional new information for this paper.

7. Are the claims appropriately discussed in the context of previous literature? For the most part yes, but not fully for the previous multi-channel seismic reflection work. Please see explanations in comment 2.

8. Is the manuscript clearly written? If not, how could it be made more accessible? Yes, the manuscript is written clearly and is a pleasure to read.

9. Could the manuscript be shortened to aid communication of the most important findings? Not taking into account the rules for manuscript length set by the journal, because I did not look into such details, I suggest no shortening for the manuscript. However, if the manuscript is considered too long by the journal, I would suggest cutting down on some parts of the methodology because most of the results were produced as part of earlier publications and do not need to be repeated.

10. Have the authors done themselves justice without overselling their claims? It appears to me that the manuscript mildly overstates the novelty of the interpretations carried out. But this is easily fixable. Please see comment 2.

11. Have they been fair in their treatment of previous literature? Please see comment 2 and further text with specific comments to address.

We have included additional discussion and citations of previous seismic reflection work and have also added a new paragraph that incorporates previous geological work on the nature of subduction plate boundaries.

12. Have they provided sufficient methodological detail that the experiments could be reproduced? My feeling is that there is perhaps more than enough detail but clarity of presented information can be improved. This particularly because it is hard to know what are the new results the authors produced for this paper and what was done for previous publications. Clarifications are needed on this in the revised manuscript. For example, was the LFE relocation and related 3D tomography in the Methods section done exclusively for this manuscript or was it done and described in an earlier another publication, like for most of the other seismological results presented. There also is a lot of information (a whole section) on the resistivity work, all of which seems published in a number of earlier papers. I don't think that any of that takes away from the significance of the work the authors have carried out, but I would like to see an explicit description of which results are new and done for this work and which are not and are used and described in earlier work and publications.

We now make clear that the double difference tomography and earthquake/LFE relocation was previously carried out, and has been published. The magnetotelluric data have been previously inverted for resistivity by two groups, and published. The resistivity model we present, however, is a new inversion carried out for this study using a subset of data from a longer profile across the Canadian Cordillera. The new model is generally similar to the earlier work, but we also present new model assessment tests for southern Vancouver Island that allow the reader to assess the quality of the resistivity result, which we now correlate with seismic reflection data. These points should be much clearer in the revised manuscript.

Other points to address as requested by this reviewer:

1. I would like to see the Abstract rewritten to focus more on what is new in this manuscript. As is, the Abstract starts with summary description of previous results/interpretations, and then moves to new results done for this manuscript, which are LFE relocations and a new 3D velocity model. This is followed by the classic “Here we show...”, all as one would expect. However, the Abstract then veers off the expected path as it describes magnetotelluric results, which I think we know from earlier papers. To me what is new in this manuscript is simple, its about (1) where the various sources of data and results point in terms of the location of the megathrust and (2) where the same data point to in terms of the deformation distribution, from focused at shallower depths to distributed at greater depth but now narrowed down to about the lower half (~3 km) of the thick reflection band. I think that being more direct and bold with their interpretation results instead of leaving the reader to work hard to figure out what the important finds are would make a much better Abstract. As is, the Abstract includes even numbers and units, such as the 30-80 Ohm-meter electrical conductor. This does not tell the reader much as one does not know how does that compare to layers above and below in terms of numbers. Nevertheless, I have no objections to keeping the Abstract as is but I do think that the current one is not the most effective at conveying the key important finds of their work but is instead convoluted.

As suggested we have completely rewritten and focussed the abstract, removing the detail on the electrical results. We present our two alternative models of the plate boundary, rather than just the vertically distributed shear zone in the lower part of the reflectors, because we cannot discriminate between the two on the basis of the available data. We have not, however, included any statement in the abstract about the landward thickening of the seismic reflectors from below the continental shelf to below Vancouver Island, because we do not deal with this issue in our paper and there is limited space.

2. Lines 12-16 and 52-56. The manuscript explains at various places, starting from the Abstract, that the oceanic crust is considered to be overpressured in many interpretations owing to fluid trapped beneath an impermeable seal along the overlying inter-plate boundary. This agreement among what appear to be many interpretations is, perhaps, not unexpected considering that, best to my knowledge, none of these interpretations actually study and consider the hydration of the incoming oceanic crust/plate but rather take into account some generic model of oceanic crust/plate. I don't think that such studies are effective because the amount of water in pores and chemically bound in the subducting oceanic lithosphere varies by orders of magnitude from one subduction zone to another. Cascadia subduction zone has the globally best constraints on the water content of the incoming oceanic lithosphere that are provided in a recent Nature Geoscience paper by Canales et al. (2017). This paper constrains the amount of pore and chemically bound water layer by layer (sediments, layer 2A, layer 2B, Layer 3 and uppermost mantle). We know that the Juan de Fuca is at the extreme dry end of subducting plates. There is a large amount of text in the Calvert et al. manuscript that explains the results from previous work. Some of it is needed because this is an integration paper. However, I fear that integrating all these results about deep and complex subduction structures that are known to have significant uncertainty and low resolution (note that there basically are no uncertainties given in the manuscript, nor any resolution tests) will not give the reader high confidence in the presented interpretation, which I actually think is well done. Why not integrate the detailed results on the hydration of the subducting Juan de Fuca plate and tell the reader where the water is coming from (which lithospheric layer) and how much water one could expect to have released and where, and how does that agree with what is observed in the resistivity and other results? The authors have a real opportunity to bring subduction zone interpretation to a new level and

it would be a shame to miss this opportunity. Of course, the authors can argue that this is out of the scope of their paper. However, at minimum, they need to put their interpretations in the context of the existing Juan de Fuca plate hydration knowledge and there is a lot of work done and published in the last 10 years.

We now cite the paper by Canales et al. (2017), and note that the fluid input to the northern Cascadia subduction zone is largely transported in the upper igneous oceanic crust and the sedimentary section. However, quantitative balancing of the fluid input to the subduction zone with the various outputs, whether released into the forearc or through the arc volcanoes is a complex question. The work by Fagereng et al. (2018) argues that fluid carried down by the igneous oceanic crust is released in relatively short duration pulses at 20-25 km and 38-40 km depth due to the breakdown of lawsonite (blueschists) and chlorite respectively as the plate descends. In other studies, which include shear heating on the plate interface, e.g. Gao and Wang (2014), the oceanic crust is predicted to be too warm for formation of lawsonite (Hacker et al., 2003). We have not detected in the tomographic velocity model the high P wave velocities of 7.5-7.8 km/s in the oceanic slab that could indicate lawsonite blueschists at 1 GPa. The velocities of 6.9-7.2 km/s are consistent with greenschist basalt and amphibolite, perhaps some epidote blueschist. So there seems to be significant disagreement between studies of metamorphic processes taking place in a warm subducting slab. Blueschists can carry a lot of water to depth, so uncertainty over their formation leads to significant uncertainty in where water is released from the igneous slab. In short, metamorphism is important and has been a field of active study by many researchers over the last few decades, but it is really beyond the scope of our study. Electrical resistivity might be able to place an upper bound on the amount of water present above the subducting slab if the thickness of the conducting layer is known, though the pore geometry introduces large uncertainties, and this was examined by Hyndman (1988). However, we do not believe that we can constrain with the available MT data the thickness of the conductive layer, which is a prerequisite for this calculation; there is always some uncertainty in the resistivity-thickness product. In summary, our discussion of fluid migration is presented qualitatively rather than on a quantitative level, which requires more clarity on the temperature and state of metamorphism in the subducting plate than currently available. The modelling study of Fagereng et al. (2018) seems to be the most quantitative result to date, but its prediction of lawsonite is not supported by other researchers in the field, who we consulted.

3. Lines 48-56 and 137-143, etc. Looking at the ideas and references presented in this manuscript, as well as from my reading of other earlier papers, I note that the various interpretations on the role of the subducting oceanic crust and lithosphere appear to lack in-depth information on oceanic crustal structure. These interpretations are challenging for me to understand as they lump the oceanic crust into upper and lower and then treat the upper crust as a homogeneous body. However, the upper crust has two layers, A and B, that are vastly different in terms of rock type, structure, porosity, anisotropy, alteration minerals filling in pore space, mechanical properties, etc. Layer 2A is made of basaltic extrusive rocks that can be represented by subhorizontally layered flows that were recently beautifully imaged (see Arnulf et al., 2014 Geology paper). Layer 2A can have a structure that has subhorizontal surfaces of mechanical weakness that could shear after subduction at great depth. However, layer 2B, which represents say 80-90% of the upper crust, is composed of vertical diabase dykes and, as such, should show greatest mechanical resistance to shearing in horizontal or subhorizontal direction of all the oceanic or continental lithospheric layers. I just simply cannot see a mechanism that would make it form horizontal or subhorizontal shear zones that exhibit reflectivity in that direction. Here I am discussing the subduction depths/pressures/temperatures targeted in the submitted manuscript, which

seem to be too shallow/small/low for significant metamorphic processes in the oceanic lithosphere to take place. So, in summary, including layer 2A into the imaged and interpreted shear zone is acceptable from my perspective but including layer 2B is not. However, including layer 2A adds only a few hundred meters to the shear zone and cannot be significant for formation of a shear zone that is 6 km or so thick. I suggest removing or modifying Figure 4c interpretation. If the authors still want to include the upper oceanic crust into the horizontal/subhorizontal shear zone that represents the plate interface area exhibiting distributed deformation, then they need to explain both (1) why would the oceanic crust in general start shearing horizontally/subhorizontally before the lower continental crust, especially considering that it is the lower continental crust that should be much more quartz rich, and (2) how can layer 2B, which is made of vertical diabase dykes, a mechanically strong material with vertical surfaces of preferential weakness, shear horizontally/subhorizontally. Therefore, it appears to me that the submitted manuscript requires some reevaluation of the earlier work on receiver function interpretation, all of which focuses on the LVZ as being due to part of the oceanic crust.

This is a good point. We now cite literature on exhumed plate boundary zones, and note that most basaltic sections are 300-500 m thick, i.e. corresponding to Layer 2A, and that stripping the deeper sheeted dike section is unlikely, and we no longer explicitly refer to these seismically defined layers. In exhumed plate boundaries, imbricated basaltic sections are repeated, implying that slivers of basaltic crust have been stripped and then stacked in a duplex structure. We have modified the figure of the plate boundary models (formerly Figure 4, now Figure 5) to include only thinner sections of upper oceanic crust. As an aside, we note that the lower crust beneath Vancouver Island is significantly more mafic than typical continental crust, because it corresponds to the accreted oceanic Wrangellia arc terrane. Please see the comments above regarding the significant metamorphism that does occur in the subducting igneous oceanic crust.

4. Lines 63-64. "LFEs lie at the base of a regionally extensive shear zone". Does this mean below it or within its lowermost part? Would be good to be specific because when reading this it seemed to me inconsistent with what is written in the Abstract on lines 13-16. Please be exact throughout the text. At the same time, this could be just me picking up on something that is not necessarily a problem.

We have clarified the text to state that the LFEs occur immediately below the reflective shear zone, but note that given the uncertainties in LFE locations it is possible some events could occur in the deepest reflectors, and we now clearly state this point. We have also included another short reflection section closer to the POLARIS profile (new Fig. 3), showing the LFEs located 0-3 km below the deepest reflectors.

5. Lines 118-122 and elsewhere. In his research, Calvert et al. are relying on crustal reflection sections that are extremely coherence filtered, more than I have seen anywhere else except in their earlier work. The extreme coherency filtering makes these reflection sections look artificial to me and make me question the interpretations derived from them. Could the authors provide in supplementary materials or in their rebuttal a comparison supplementary figure that shows the reflection section from Figure 3 together with its equivalent that is not coherency filtered and moderately coherency filtered so that we can see if the ~6-10 km thickness of the wide reflection band is observable in these sections as well. The 10 km thickness and lateral extent of >10 km (as stated in line 182) seems to me to be possibly too large.

]We have provided in the supplementary material an explanation of the development of the image of the seismic section around southern Vancouver Island. The reflections and their thickness can now also be evaluated on the unmigrated stack section. We note wave-equation based migration, which improves spatial resolution in theory, can produce large lateral smearing of the data, and therefore give

an unrealistic impression of the lateral continuity of reflectors, which is why we have employed a different algorithm that migrates individual reflector segments according to their apparent dip. We now characterize the lateral continuity as “up to 10 km”.

6. Lines 133-135. Interpretations are made here but also elsewhere in the manuscript based on the values of various properties derived from the collected data such as P and S wave velocities, change in these velocities from the starting model, resistivity, etc. But nowhere in the manuscript I did see any information on the uncertainty these number have. My guess is that the resolution is low and uncertainties high for the ~7 km/s velocity mentioned here for the E reflections area discussed. Would it be possible to give error bounds doe the ~7 km/s velocity and any resolution tests for these velocity models? This would make the conclusions that these are not metasediments, which I agree, stronger.

We have now included in the Supplementary Information checkerboard and dipping layer resolution tests that were published with the original tomographic velocity models. We have also added a brief comment noting that the error implied by the checkerboard test does not alter the interpretation of mafic lithology within the E reflectors beneath Vancouver Island. The increase in seismic velocity along strike at ~35 km depth from 6.6 km/s below the Olympic Peninsula to 7.0 km/s beneath the POLARIS profile can be seen in a previously published tomographic P wave velocity model (Preston, 2003; Calvert et al. 2011), and is also visible in the TomoDD velocity models we present in the current paper. We have not included this figure, as it is not a new result.

In summary, this manuscript comes from an excellent and productive team of researchers. Their presented work is very interesting and deserves to be published in Nature Communications after a revision.

Mladen Nedimović

Reviewer #2 (Remarks to the Author):

The manuscript presents new earthquake relocation results in the context of multiple structural seismology images and a magnetotelluric image of the northern Cascadia subduction zone. The new results, particularly the low-frequency earthquake hypocenters, are used to motivate a new synthesis of plate boundary structure from imaging and earthquake studies focused on a transect beneath southern Vancouver Island. While there is already a sequence of papers making inferences based upon some of the results presented here, the text is right to point out that there still is not agreement on the location of the plate interface with respect to the commonly imaged low-velocity zones, broadband earthquakes, and low-frequency earthquakes. This manuscript makes a compelling attempt weave together all these lines evidence with consistent modeling of P and S velocity structure, reflection/conversion imaging, earthquake locations, and conductivity. The manuscript’s integrative results and interpretation rise to the level of a significant new contribution despite the extent of prior research on the subduction interface in this region.

The manuscript is clearly written and the supplemental description of method is helpful. Hypothesis-driven resolution tests seem to be the only potential ingredient that is missing. It would be helpful to know more about what level of resolution of 3D hypocenters and LVZ (and conductivity) boundaries is feasible given the data used here (source-receiver geometry, frequency content, etc.). I realize this is not

a small task given the breadth of data types considered here, but it would be valuable to have greater treatment of the resolution limits in this manuscript rather than just references to other studies. It would strengthen the interpretations if synthetic data for a hypothetical model were inverted for seismic structure & hypocenters and conductivity structure.

We now include the checkerboard resolution analysis carried out for the 3D seismic velocity inversion, and a test of the ability of the geometry of sources and receivers used in the study to recover the V_p/V_s anomaly of a dipping layer. As noted above, and in the manuscript, this assessment work was previously published with the velocity models. We also include examples of the inversion of the magnetotelluric data that demonstrate the degree of non-uniqueness in the resistivity model derived from the magnetotelluric data, together with a modelling study that shows that the landward dipping conductor terminates, and probably rises into the North American plate at its landward end.

Reviewer #3 (Remarks to the Author):

General comments

I am not over familiar with the literature for this margin, and this is probably a central reason for me finding this paper a bit difficult to read. So from my perspective, the reader needs a bit more help to: understand the main goals and principal findings of the study, distinguish between what's already known (previous geophysical results and models) and what's new in the manuscript. The aims and objectives of the current study are not specifically stated anywhere. I agree that these can be partially inferred from the current text, but the paper will have much more impact if the story is better told. On the first read of the paper, I initially thought that the P and S wave tomographic models, relocation of the LFEs, and MT results were all new, but the citing in lines 60-62 and 82-83 suggests they have already been published in reference 20, 8, 26 and 27. And when I looked at the figures, I wondered whether the aim was to test between the different models shown in Fig. 4, but there doesn't seem to be any conclusion about which of these best explains the available data. Then I wondered whether Figure 5 was the principal result – which it might be – but this figure isn't actually cited in the text.

We agree that the original introduction to the paper was insufficiently detailed and not very explicit about the objectives of our paper, one of which is to reconcile the fundamental inconsistencies between existing teleseismic and active source studies. For example, our results imply that previous interpretations of metamorphism in the subducting slab from teleseismic observations of the LVZ should be reconsidered, because the LVZ lies in the overriding plate, or at least forms part of the plate boundary. We have rewritten, clarified and lengthened the Introduction, making explicit our integration of multiple previously published studies. We present two models that satisfy the available geophysical constraints, but cannot really state that one model is clearly superior to the other, though we note a preference for the thicker plate boundary. We present Figure 5 (now Figure 6) to summarise the relationships between the structures inferred from the different geophysical methods... and now cite it in the text.

Although Nature Communications papers are formatted slightly differently, they do recommend having a 'Results' section. Papers (the few that I skimmed) do seem to also have a classic introduction, where the 'state of play' is described just before the results section, and there is a clear statement of the aims and objectives of the current study, how these aims are achieved, and what results have been achieved.

So I think the paper could be improved with some re-structuring, with clearer statements about what is known and what is new, include a statement of the aims and objectives of the study, and have a separate results section. Perhaps the aim is to distinguish between the three alternative models of the boundary (fig. 4), or better locate the geophysical properties across the inter-plate boundary, and the approach is to use existing data and generate some better geophysical plots (i.e. figures 2 and 3)?

The overall structure of the paper has been reorganised to conform with the requirements of Nature Communications, with Introduction, Results, and Methods sections. In the rewriting of the Introduction, we now summarise more of the previous seismic reflection and receiver function studies, and clarify that we are integrating in our study a variety of earlier results. Even with this integration, we note that both presented models of the plate boundary are compatible with the data, and we have made this clear, though we express a preference for the distributed boundary zone.

In summary, as currently written, it is not clear to me whether there is sufficient scientific advancement in the paper to merit publication in Nature Communications.

Please see initial introductory comments on page 1.

Detailed comments

In lines 27-68 and 170-223, the text jumps about a bit too much with not enough linking between concepts, contains sentences that appear to be a distraction from the main story, and the text is a bit too passive (over usage of 'can' and 'may').

The Introduction originally presented in lines 27-68 was disjointed, and has been extensively revised, hopefully making it much clearer. In lines 170-223, which corresponds to the section on fluids and the inter-plate boundary, we have introduced further section headings and made some changes for clarity, but we have tried to maintain the general development that elicited positive comments on the writing from Reviewer 1.

In lines 66-68 "Consequently, features previously associated with the subducting oceanic plate either lie within the overriding plate or form part of a plate boundary zone a few km thick", it is not clear exactly what features are being referred to or why it is important. The linking between statements like this one and the text below is poor – which makes the story difficult for the reader to follow.

We make explicit that the key feature is the region of elevated Poisson's ratio, and place this text in a completely revised final paragraph of the Introduction.

Line 89 – is the identification of the ULVZ new, or is it already published in reference 17, or is reference 17 a citation for the adopted methodology.

The region of high Poisson's ratio was previously identified in the cited reference, but we introduce the term ULVZ here to simplify the subsequent discussion in our paper.

Line 106 Can tomographic inversions make associations?

This wording on line 106 has been revised.

Line 121 – I am going to assume that I have permission to moan here. The interpretation that the E reflectors are formed by shearing is an interpretation – not one that I personally think is correct. The E

reflections look like normal layered lower continental crust, into which faults in the upper continental crustal faults merge, as expected for a brittle to ductile transition. We don't know what causes lower crustal reflectivity (there is no consensus), but it is much easier to generate high-amplitude reflections with lithological differences, such as intrusion of mafic igneous rocks into the lower continental crust (which would be consistent with an average velocity of 7 km/s). Lithological layering (as the cause of the reflectivity) does not preclude the occurrence of shearing after the formation of the reflectors. And intrusions would be easier to envisage as (semi) impermeable barriers. Is there a good reason to exclude this? Anyway, I'd like to request that the revised text is more equivocal about the cause of the reflectivity.

We have now been more equivocal with regard to the possible origins of the seismic reflections mentioned on line 121, and have cited previous studies in the context of subduction zones where the reflections can be traced to the megathrust fault offshore, but we have not included magmatic intrusion as a possible explanation. It is certainly very striking that the thick sequence of reflections in the lower forearc crust of Cascadia appears similar to the subhorizontal reflections observed in the lower continental crust, for example around the UK and in the US Basin and Range region, where intrusion of mafic igneous rocks would be a favoured interpretation. In Cascadia, however, there is no evidence of large-scale magmatic activity in the forearc, which is uncommon in subduction zones world-wide, being associated mostly with subduction of oceanic ridges and the generation of slab windows. In Cascadia, the shallowly landward dipping reflections can be followed on multiple seismic lines to depths of 50 km or more (Calvert et al 2006), which is well below the continental Moho at 35 km (Zelt et al., JGR, 1996), precluding an origin related to intrusion into the lower continental crust. The seismic reflectors are subparallel and above the inslab seismicity, where observed, and given their connection to the offshore inter-plate boundary, the most logical interpretation is that they arise in rocks sheared close to the plate boundary, but we are now more equivocal about exactly how the seismic reflectivity arises. Whether the reflectors are a thick zone of active shearing (Nedimovic et al 2003) or have been sheared and underplated to the overriding continent is open to interpretation, but we now cite in the paper geological studies of exhumed plate boundaries that support this deformation-related interpretation. In the bigger picture, it appears that oceanic arcs, which are built by magmatic intrusion over millions of years are mostly non-reflective (at normal incidence), so it seems that magmatic intrusion on its own, in general, is not sufficient to produce a thick pervasive sequence of high amplitude reflections (as opposed to individual sills), and that a favorable stress regime facilitated by a ductile rheology may well be required, as suggested by the reviewer. The lower crust will also be weakened by the intrusion of significant volumes of melt, so elevated lateral tectonic stresses, leading to extension or compression of the crust, could result in synchronous melt intrusion and shearing, e.g. migmatitic gneisses, which may explain much of the seismic reflectivity observed in Archean terranes. However, it is certainly true that we still do not know enough about the various causes of deep seismic reflectivity in different tectonic settings.

The scenarios shown in Figure 4 are first noted in lines 124 and 127, and then later they are referred to as model 1 and model 2 (lines 144-147). It would help the reader if there was a clear statement (on their first mention) that there are three models that may explain the data, and for the features in each of the proposed models to be explained and discussed. Is there a preferred model? This is not clear from lines 179-180.

We have reorganised and rewritten the text that introduces the models of the inter-plate boundary, removing one of the options, and have simplified the figure to avoid confusion. Both models are

compatible with the available data, but we have now expressed a tentative preference for a thicker boundary since active shearing may be better able to maintain the fluid porosity and overpressures.

Resistivity model (image suggests something that is well-resolved). MT data have a relatively low resolution and we can expect recovered resistivity anomalies to be a very smooth version of reality with not very good anomaly thickness and depth control. With this in mind, the use of the words “consistent with” is fine, but stronger assertions should be avoided.

We completely accept the reviewer’s comments on the smoothness and resolution of the resistivity model derived from magnetotelluric data, and have now qualified our characterization of the model and our interpretation of its relation to the seismic reflectors.

The red zone looks to be between CDPs 4400 and 5100, be relatively flat or slightly curved at its base, and the top of the red zone is curved upwards towards the surface in the landward direction. Is there a good reason to be confident about the dip and depth of the yellow/orange resistivities, to be able to say it is approximately coincident with the ULVZ?

In the Supplementary Information we provide multiple examples of the resistivity inversion of the field data, and examples of the inversion of synthetic data calculated for specific subsurface models. We use these results to characterize the uncertainty in what can be interpreted from the resistivity model, and now note that some inversions permit the conductor to lie below the reflectors at their western end.

Line 159, the resistivity and E reflectors don’t correlate that well. The red zone correlates with the E reflectors at about CDP 4600, but at higher CDPs the red zone is above the E reflectors.

We have expanded this description to better characterize the resistivity anomaly, and now note that it rises eastward, an interpretation which is supported by the new synthetic inversions in the Supplementary Information.

Line166 Sentence “Broadly consistent with” needs re-writing. Consistent with the interpretation that there is a...

We have rewritten the phrase “broadly consistent”.

Line 185-186. This statement is too strong – the electrical conductor is not coincident with the E reflectors.

We have removed our assertion that the conductor is coincident with the E reflectors.

Figure 5. This figure isn’t cited.

The summary figure is now cited as Figure 6.

Prof Joanna Morgan, Imperial College London

REVIEWERS' COMMENTS:

Reviewer #1 (Remarks to the Author):

I completed my review of the revised manuscript entitled "The slow-slip inter-plate boundary in the northern Cascadia subduction zone" by Dr. Calvert and co-authors (manuscript number NCOMMS-20-05972A), which has been submitted for publication in Nature Communications. I recommend that the work by Calvert et al. is published in Nature Communications as is. My comments for this round of review are short because the authors have surpassed my expectations with their revision. They have done a great job at cutting out less important details and focusing on the big picture by clarifying the key aspects of their integrative interpretation while responding to all comments by the reviewers. I look forward to seeing this work published.

Mladen Nedimović

Reviewer #2 (Remarks to the Author):

The authors responded thoroughly to the initial reviews strengthening the communication of the new integrative analysis and better placing the interpretations in the context of the prior literature. The revised text clarifies how prior results are reconsidered using a consistent velocity model and better identifies the limiting factors that previously motivated inconsistent structural interpretations from different imaging methods. I think the manuscript is essentially suitable for publication and is a good fit for the journal. Two very minor comments are included below as they may help polish the already strong communication of the revised manuscript.

Abstract. It might help to specifically note that the zone of high PR is shown to be within the base of the upper plate. Readers should be able to follow the current wording to reach that conclusion, but such a minor modification might help highlight one of the key interpretations of this study that differs from some prior influential imaging studies.

Figure 2's caption should note the labeling of LFEs and normal earthquakes.

Brandon Schmandt

Response to Reviewers

Reviewer #1 (Remarks to the Author):

I completed my review of the revised manuscript entitled “The slow-slip inter-plate boundary in the northern Cascadia subduction zone” by Dr. Calvert and co-authors (manuscript number NCOMMS-20-05972A), which has been submitted for publication in Nature Communications. I recommend that the work by Calvert et al. is published in Nature Communications as is. My comments for this round of review are short because the authors have surpassed my expectations with their revision. They have done a great job at cutting out less important details and focusing on the big picture by clarifying the key aspects of their integrative interpretation while responding to all comments by the reviewers. I look forward to seeing this work published.

Mladen Nedimović

We thank Dr Nedimović for the considerable effort he put into reviewing our manuscript.

Reviewer #2 (Remarks to the Author):

The authors responded thoroughly to the initial reviews strengthening the communication of the new integrative analysis and better placing the interpretations in the context of the prior literature. The revised text clarifies how prior results are reconsidered using a consistent velocity model and better identifies the limiting factors that previously motivated inconsistent structural interpretations from different imaging methods. I think the manuscript is essentially suitable for publication and is a good fit for the journal. Two very minor comments are included below as they may help polish the already strong communication of the revised manuscript.

Abstract. It might help to specifically note that the zone of high PR is shown to be within the base of the upper plate. Readers should be able to follow the current wording to reach that conclusion, but such a minor modification might help highlight one of the key interpretations of this study that differs from some prior influential imaging studies.

We have slightly modified the abstract to improve its clarity, but we have not specifically included the above point, because it doesn't seem to allow for the possibility that the region of high Poisson's ratio could be a vertically distributed plate boundary zone, i.e. Model 2 in Figure 5b, which we consider to be below the upper plate (and above the lower plate). So to be specific we do not view a vertically distributed plate boundary as being part of the upper plate. It is difficult to be more assertive in the abstract, because we want to make it clear that we are allowing for each of the two models, as described in Figure 5.

Figure 2's caption should note the labeling of LFEs and normal earthquakes.

The caption to Figure 2 has been modified accordingly.

Brandon Schmandt

We thank Dr Schmandt for the considerable effort he put into reviewing our manuscript.